# Human *MLH1/3* variants causing aneuploidy, pregnancy loss, and premature reproductive aging

Priti Singh[1,2], Robert Fragoza[3,4], Cecilia S. Blengini [5], Tina N. Tran[1], Gianno Pannafino[4], Najla Al-Sweel[4], Kerry J. Schimenti[1], Karen Schindler[5], Eric A. Alani[4], Haiyuan Yu [3,6] & John C. Schimenti [1,4 ✉]

Embryonic aneuploidy from mis-segregation of chromosomes during meiosis causes pregnancy loss. Proper disjunction of homologous chromosomes requires the mismatch repair (MMR) genes *MLH1* and *MLH3*, essential in mice for fertility. Variants in these genes can increase colorectal cancer risk, yet the reproductive impacts are unclear. To determine if *MLH1/3* single nucleotide polymorphisms (SNPs) in human populations could cause reproductive abnormalities, we use computational predictions, yeast two-hybrid assays, and MMR and recombination assays in yeast, selecting nine *MLH1* and *MLH3* variants to model in mice via genome editing. We identify seven alleles causing reproductive defects in mice including female subfertility and male infertility. Remarkably, in females these alleles cause age-dependent decreases in litter size and increased embryo resorption, likely a consequence of fewer chiasmata that increase univalents at meiotic metaphase I. Our data suggest that hypomorphic alleles of meiotic recombination genes can predispose females to increased incidence of pregnancy loss from gamete aneuploidy.

[1] Dept of Biomedical Sciences, Cornell University College of Veterinary Medicine, Ithaca, NY, USA. [2] Preclinical Modeling Core Lab, Fred Hutchinson Cancer Research Center, Seattle, WA, USA. [3] Weill Institute for Cell and Molecular Biology, Cornell University, Ithaca, NY, USA. [4] Department of Molecular Biology and Genetics, Cornell University, Ithaca, NY, USA. [5] Rutgers University, Dept. of Genetics, Piscataway, NJ, USA. [6] Department of Computational Biology, Cornell University, Ithaca, NY, USA. ✉email: jcs92@cornell.edu

The essential function of meiotic recombination is to drive pairing of homologous chromosomes, ensuring proper disjunction of homologs to daughter cells at the first meiotic division. Failure of a chromosome pair to undergo at least one crossover per chromosome arm predisposes to formation of aneuploid gametes that, if fertilized, leads to pregnancy loss. Approximately 50–70% of early miscarriages are associated with chromosome abnormalities, with most having aneuploidy, primarily trisomies[1,2]. It is well recognized that the incidence of aneuploidy in abortuses increases with female age and has been hypothesized to be attributable to non-genetic factors such as decay of chromosome cohesion[3,4]. However, there are indications that parental genetic factors can predispose to generation of aneuploid conceptuses, and may contribute to cases of recurrent pregnancy loss (RPL; defined as three or more consecutive miscarriages)[5,6]. For example, much attention has focused on the potential effects of variants/mutations in the synaptonemal complex protein SYCP3 in human RPL[7,8], because SYCP3-deficient female mice experience loss of embryos resulting from aneuploid oocytes[9]. However, it is unclear whether, or to what extent, mutations or deleterious variants in SYCP3 or other genes cause miscarriages or RPL.

Chromosome mis-segregation during meiosis can occur at Meiosis I (MI) or Meiosis II (MII), though a lack of recombination on extra/missing chromosomes has been associated with inefficient crossover maturation before MI[10]. The resulting gametes, if fertilized, can lead to aneuploid embryos that are almost universally incompatible with viability, thus causing miscarriage. Several genes are required for crossover (CO) recombination (and thus fertility) in mice, including the mismatch repair (MMR) proteins MLH1 and MLH3, which form a heterodimeric endonuclease required to resolve double Holliday junction recombination intermediates[11–19]. Null alleles of Mlh1 or Mlh3 cause meiotic arrest and sterility in mice because they are needed for ~90% of all crossovers (those known as interference-dependent Class I crossovers)[11,20–25]. The absence of COs leads to univalent chromosomes that fail to align properly while attached to microtubules at the metaphase plate, thus triggering the spindle assembly checkpoint (SAC) and blocking anaphase progression[26]. However, recombination defects affecting bivalent formation of only one or a few chromosome pairs is inefficient at triggering the SAC, allowing such gametes to survive and thus predisposing to aneuploidy[27–30].

We hypothesized that human population variants in MLH1/3 might cause defects in chromosome segregation during MI or MII, leading to infertility, miscarriage, and/or developmental defects. To test this and gain a comprehensive understanding of the functional impact of such variants, we systematically evaluated human missense single nucleotide polymorphisms (SNPs) curated in the ExAC and (subsequently) gnomAD databases[31,32]. Minor alleles predicted to be deleterious in silico by various algorithms, and which deviated from normal function in biochemical assays or genetic assays in yeast, were selected for modeling in mice. Most of the mutant mouse models were fertile but exhibited crossing over defects that led to age-related decreases in female fecundity, elevated aneuploidy, and consequent increases in pregnancy loss. These results have implications especially for older couples, or those experiencing RPL, who may be at increased risk for having conceptuses or children with trisomies as a result of bearing variants in these recombination genes.

## Results

### Strategy and selection of candidate pathogenic MUTL homolog variants.
To test the possibility that segregating infertility variants of MLH1 or MLH3 exist in human populations, we identified potentially pathogenic missense SNPs in these genes using methods and criteria outlined in Fig. 1a, including: (1) functional prediction algorithms scoring the variant as deleterious; (2) has an allele frequency (AF) within the range of 0.02 to 2% in any gnomAD-listed population; (3) disruptive to binary protein-protein-interactions in yeast two-hybrid (Y2H) assay; or (4) causes an MMR and/or recombination phenotype in baker's yeast (S. cerevisiae) when the orthologous amino acid is altered. These steps were used to prioritize a subset of MLH1 and MLH3 variants for subsequent modeling in mice. In particular, steps 3 and 4 were explored as a means to potentially improve the success rate of identifying functionally consequential infertility variants for mouse modeling, compared to our previous studies that utilized only computational predictions[33,34].

Disease-associated mutations in MLH1 often function molecularly by disrupting corresponding protein-protein interactions[35,36]. As such, we searched for all potentially functional MLH1 missense variants that matched our defined 0.02% < AF < 2% criteria range. These MLH1 variants were introduced into Y2H bait and prey vectors, then tested for whether they disrupted known interactions with 8 proteins that were available as full-length clones in the hORFeome v8.1 or the Mammalian Genome Collection (MGC). Overall, 13 MLH1 variants were tested by Y2H across 68 total SNP-interaction pairs. We then further filtered these variants for SNPs that scored as deleterious across multiple functional prediction algorithms. This resulted in five MLH1 predicted deleterious variants that disrupted 27 out of 37 (73%) of their corresponding SNP-interactions pairs (Table 1; Supplementary Data 1).

For MLH3, we modeled eleven human SNPs in yeast by constructing strains bearing the cognate amino acid (AA) changes in the endogenous MLH3 gene, then measuring MMR using a lys2 frameshift reversion assay[37]. Most of the AAs are located in conserved endonuclease and ATP binding domains (Table 2; Supplementary Fig. 1). To measure meiotic crossing over, these strains were mated to an mlh3Δ strain so that the diploids contained fluorescence markers separated by ~20 cM on chromosome VIII[38]. The diploids were sporulated to identify, in complete tetrads, presence or absence of crossovers (Supplementary Fig. 2). Three of the MLH3 SNPs (rs781779034, rs138006166 and rs781739661) had null-like or intermediate phenotypes for MMR, and null-like crossing over defects (Tables 1 and 2; Supplementary Table 1). Additionally, all three were classified as deleterious by functional prediction algorithms (Table 1). Overall, nine of the eleven alleles modeled in yeast conferred null-like or intermediate phenotypes in either or both tested metrics (Table 2).

### Mouse modeling of nsSNPs and reproductive phenotypes.
Based on the bioinformatic, biochemical and genetic data described above, we initially modeled five MLH1 and three MLH3 nsSNPs in mice via CRISPR/Cas9-mediated genome-editing (Table 1, Supplementary Fig. 2). These eight mutant mouse lines with 'humanized' alleles were phenotyped in several ways, beginning with male reproductive parameters. Two mutants had obvious phenotypes. $Mlh3^{R1230H/R1230H}$ males exhibited a ~4 fold reduction in testes size, and an absence of epididymal sperm (Fig. 1b, c). Consistent with the absence of sperm, histological sections of mutant testes revealed a lack of haploid spermatids, and the most advanced spermatocytes appeared to be at meiotic metaphase I or anaphase I (Fig. 1d), reminiscent of null animals for both Mlh1 and Mlh3[11,14,21]. The phenotypes of $Mlh1^{K618T/K618T}$ males were less severe; their testes weighed 80% of WT, and there were ~35% fewer epididymal sperm (Fig. 1b, c). Histology also revealed a substantial number of spermatocytes arrested in metaphase I (Fig. 1d). Fertility trials of homozygous mutant males

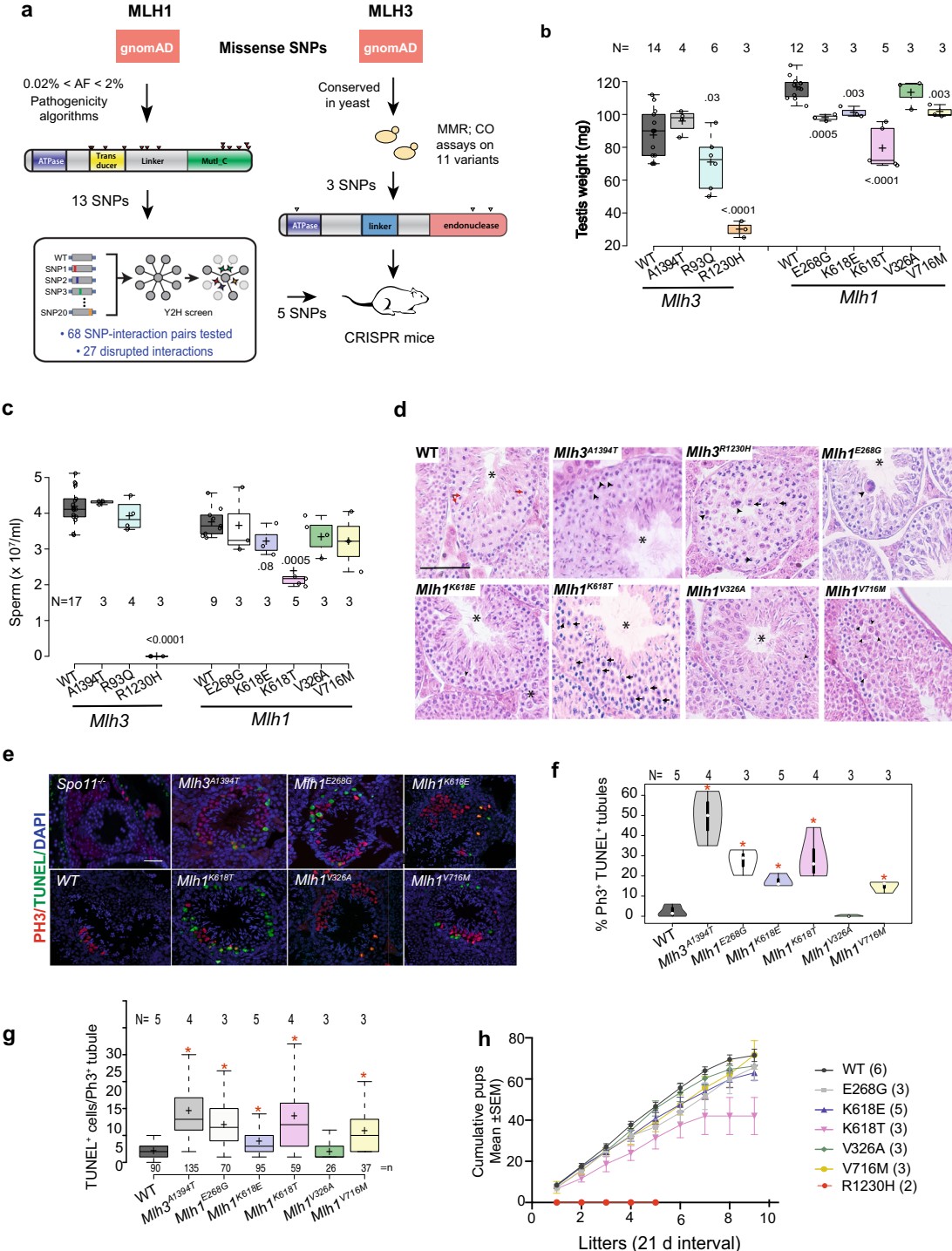

confirmed that $Mlh3^{R1230H/R1230H}$ males were completely sterile, whereas $Mlh1^{K618T/K618T}$ males were subfertile, producing 40% fewer pups over a duration 10 months of fertility trials (Fig. 1h). In contrast to $Mlh1^{K618T}$ and $Mlh3^{R1230H}$, there were no obvious reproductive defects in males homozygous for the remaining 6 alleles that were modeled. Sperm counts were normal and litter sizes sired by these mutant males were not lower than controls (Fig. 1c, h). Testis weights were also normal, though modestly lower in $Mlh3^{R93G/R93G}$ homozygotes.

Lack of COs between homologs at meiotic metaphase I in $Mlh$-null mutants causes a failure of proper alignment at the metaphase plate, leading to anaphase block[11,20–25]. Despite the absence of reproductive defects in the remaining 6 mutant alleles, testis sections revealed abnormal pyknotic-appearing metaphase spermatocytes with disorganized metaphase plates and mis-aligned chromosomes (Fig. 1d; explored in more detail below). This was more directly assessed by TUNEL staining of seminiferous tubule cross-sections positive for the metaphase marker phospho-histone H3 (Ser10)[39] which showed marked increases in apoptotic cells compared to WT (Fig. 1e–g). Confocal analysis readily revealed misaligned chromosomes at the metaphase plates of MI spermatocytes in the less severe mutants (Supplementary Fig. 3). These data suggested that the allelic variants caused severe-to-subtle defects in chromosome

**Fig. 1 Spermatogenesis defects in mutant mice. a** Overview of process for selecting human *MLH1* and *MLH3* variants for mouse modeling. gnomAD (https://gnomad.broadinstitute.org) was the source of Allele Frequencies (AF). MMR, mismatch repair; CO, crossover. The SNPs tested are listed in Table S1. **b** Testis weights of mutants. The *Mlh1* and *Mlh3* samples were taken from 6 month and 2–3 month old mice, respectively, hence the difference between WT samples. For this and panel C, numbers are p values relative to corresponding WT of that cohort; only genotypes with values <0.05 are shown. Genotype abbreviations: WT, wild-type E268G, $Mlh1^{E268G/E268G}$; K618E, $Mlh1^{K618E/K618E}$; K618T, $Mlh1^{K618T/K618T}$; V326A, $Mlh1^{V326A/V326A}$; V716M, $Mlh1^{V716M/V716M}$. A1394T, $Mlh3^{A1394T/A1394T}$; R93Q, $Mlh3^{R93Q/R93Q}$; R1230H, $Mlh3^{R1230H/R1230H}$. **c** Epididymal sperm counts. All were from 6-month old mice, except $Mlh3^{A1394T/A1394T}$ and $Mlh3^{R93G/R93G}$, which were from 3 month old animals). Controls were from pooled littermates. Genotype abbreviations are as in "b." **d** Testis histology. Examples of H&E-stained seminiferous tubule cross sections of homozygotes for the indicated genotypes. Asterisk (*) represents Stage XI–XII tubules with metaphase cells. Arrowheads represent pyknotic or multinucleated degenerating metaphase cells. Red arrows indicate normal metaphases in WT sections, while black arrows indicate abnormal metaphases with or without lagging chromosomes. Scale Bar = 50 μm. **e** Testis sections immunolabeled by phosphorylated histone H3 (PH3, red), and also TUNEL (green)-stained for apoptosis counterstained with DAPI (for DNA, blue). Mlh1/3 testes are from homozygous mutants. Scale Bar = 50 μm. Genotypes are indicated. **f, g** Quantification of TUNEL data from samples in "E." Plotted are overall percentages of TUNEL$^+$ cells (**f**) and also tubules containing >4 TUNEL$^+$ cells (**g**). This number was chosen because no WT tubule sections had >4 TUNEL$^+$ cells. There are 2 N/n values: n = Total numbers of tubules sections analyzed for each genotype, and N = number of mice (biological replicates from which the tubules came. At least >25 Ph3$^+$ tubules from each mouse/genotype were analyzed. *$p \leq 0.0003$. **h** Fertility trials of homozygous males and control littermates. *$p \leq 0.0001$. The X-axis values are in 21-day intervals, which approximates litter intervals for fertile females housed continuously with a male. Number of animals that underwent fertility trials (N) are shown in parentheses. All p-values were from Student's two-tailed t-test. Data are presented as mean values ± SEM. In **b, c, f, g**, each plot is defined as follows: Horizontal line in box = median; box width = data points between the first and third quartiles of the distribution; whiskers = minimum and maximum values; '+' sign = mean value. N represents number of biologically independent animals/genotype examined in each experiment.

**Table 1 SNPs modeled in mice.**

| Gene | SNP | Human allele | Benign/Delet. | MAF (Population) | Functional assay | Protein domain |
|---|---|---|---|---|---|---|
| MLH3 | rs28756978 | R93G | None/S,P,R,M | 0.076% (EAS) | CO- | ATPase domain |
| MLH3 | rs781739661 | R1230H | None/S,P,R,M | 0.025% (EAS) | MMR- CO- | MutL C-ter dimer; endo. domain |
| MLH3 | rs138006166 | A1394T | None/S,P,R,M | 0.039% (NFE) | MMR+/− CO- | MutL C-ter dimer; endo. domain |
| MLH1 | rs63750650 | E268G | None/S,P,R,M | 0.16% (Fin) | Y2H (3/8) | Transducer domain |
| MLH1 | rs35001569 | K618E | None/S,P,R,M | 0.57% (NFE) | Y2H (7/8) | Exo1 Interaction domain |
| MLH1 | rs63750449 | K618T | None/ S,P,R,M | 0.57% (NFE) | Y2H (8/8) | Exo1 Interaction domain |
| MLH1 | rs3550531 | K618A | None/S,P | 0.34%* (overall) | ND | Exo1 Interaction domain |
| MLH1 | rs63751049 | V326A | P**/S,R,M | 0.11% (SAS) | Y2H (4/5) | Transducer domain |
| MLH1 | rs35831931 | V716M | None/S,P,R,M | 0.20% (NFE) | Y2H(5/8) | C-terminal |

gnomAD (http://gnomad.broadinstitute.org) allele frequencies (version 2.1.1) correspond to the population with the highest allele frequency (AF).
EAS East Asian, NFE European (non-Finnish), Fin Finnish, SAS South Asian, S SIFT, P Polyphen2, REVEL R, M Mutation assessor, ND no data, MMR mismatch repair, CO meiotic crossing over phenotypes (+, wildtype, −, null), dimer dimerization, endo endonuclease, "Y2H" yeast two-hybrid disruption, with the values in parentheses indicating the fraction of known interactions with proteins disrupted by the corresponding MLH1 mutation (see details in Table S1).
*This allele was generated near the end of the study, and consists of a dinucleotide change, each corresponding to one of the nucleotides altered in the K618E and K618T alleles. REVEL, Mutation Assessor and CADD scores were not obtained for K618A because this allele is not a recognized population variant. gnomAD lists overall AF for K618A, but its NFE-specific AF is likely identical to K618E and K618T, which almost always co-occur (see text). For functional assays in yeast, see details in Table S2.
**This pathogenicity score is classified as "possibly" damaging.

**Table 2 Analysis summary of yeast equivalents to human *MLH3* SNPs for defects in crossing over and mismatch repair.**

| Variant | SNP ID | Yeast Equiv | MMR | CO | Domain |
|---|---|---|---|---|---|
| G62R | rs761501352 | T65R | + | + | ATP binding |
| R93Q | rs781779034 | R96Q | +/− | − | ATP binding |
| R93G | rs28756978 | R96G | ND | − | ATP binding |
| P183L | rs759067670 | P199L | +/− | − | ATP binding |
| N291K | rs767413852 | L313K | + | + | Conserved amino acid |
| F390I | rs61752721 | R407I | +/− | + | undefined |
| R1230C | rs746431837 | R530C | − | − | Endonuclease |
| R1230H | rs781739661 | R530H | − | − | Endonuclease |
| R1232C | rs550698696 | R532C | +/− | + | Endonuclease |
| A1394T | rs138006166 | A702T | +/− | − | Endonuclease |
| G1396W | rs368876363 | G704W | +/− | +/− | Endonuclease |

"Variant" refers to amino acid change conferred by the indicated SNP. "Yeast equiv" is the orthologous amino acid in *Saccharomyces cerevisiae*. *MMR* mismatch repair, *CO* crossing over phenotype. For these phenotypes, "+" is WT level, "−" is null, and "+/−" is intermediate. *ND* no data. See Supplementary Table 1 for the complete dataset.

alignment at the metaphase plate at MI, and/or unrepaired recombination intermediates, leading to apoptosis.

When this work began, the two K618 variants in *MLH1* (K618E and K618T) were (and are still) present in dbSNP as rs35001569 and rs63750449, respectively. These amino acid changes are caused by AAG>GAG and AAG>ACG nucleotide changes, respectively. However, after those mouse models were created and phenotype analyses were performed (including female data presented below), the gnomAD database (v2.1) was revised to indicate that these two variants co-occur (are in phase) nearly perfectly (963/973 and 963/971 occurrences, respectively), encoding a K618A AA change as a result. This explains why both variants are annotated to occur at essentially identical frequencies in populations (Table 1). We therefore generated an $Mlh1^{K618A}$ allele and found that both male and female homozygotes are fertile. Thus, this allele is either a neutral variant or has subtle defects that may be revealed with more comprehensive studies such as those described below.

**Meiotic recombination defects in mutants**. To determine if defects in crossing-over might underlie the metaphase I

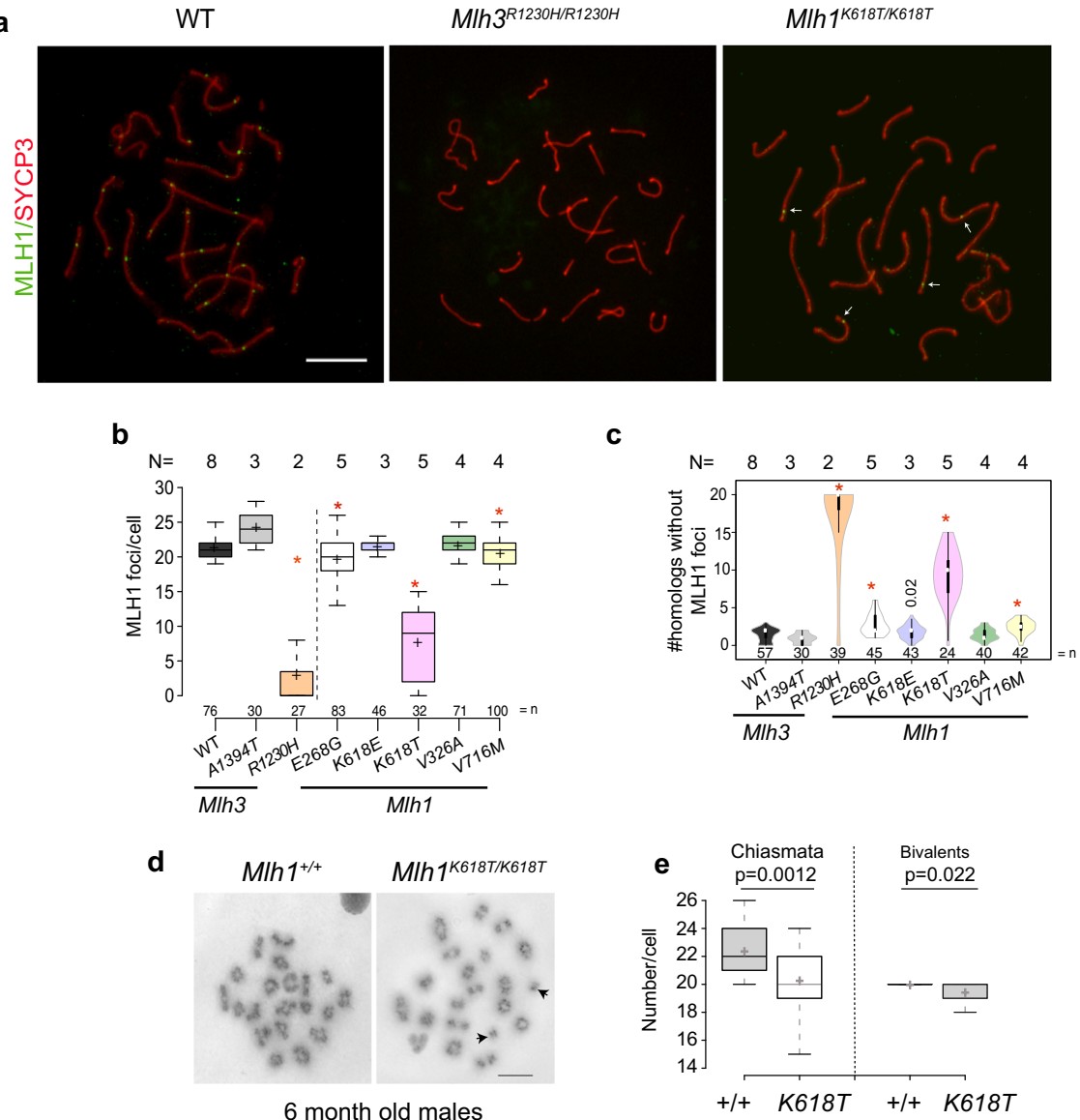

**Fig. 2 Recombination in mutant spermatocytes. a** MLH1 immunolocalization in spermatocyte nuclei. Shown are pachytene spermatocyte chromosome surface spreads from 6 month old males, immunolabeled with SYCP3 (red) and MLH1 (green). White arrows indicate MLH1. Note weak and nearly absent staining in the mutants. **b** MLH1 focus quantification in spermatocytes. *p-value <0.0001. Others do not have p-values < 0.05 for decreased foci. Over 30 cells from at least 3 animals were quantified, except for *Mlh3*[R1230H], where 27 cells from two animals were scored. **c** Violin plots illustrating the chromosome pairs in a spermatocyte nucleus lacking an MLH1 focus. White circles show the medians; box limits indicate the 25th and 75th percentiles; whiskers extend 1.5 times the interquartile range from the 25th and 75th percentiles; polygons represent density estimates of data and extend to extreme values. For mutants with more MLH-deficient pairs, *p-value <0.0001 unless otherwise indicated; comparisons to WT with p-values >0.05 are not shown. **d** Example of spermatocyte metaphase spread from the *Mlh1*[K618T] mutant and control. **e** Quantification of chiasmata and bivalents in *Mlh1*[K618T] and WT. Twenty or more cells from three animals were analyzed. Scale Bar = 5 μm. Data in **b** and **e** are plotted as follows: Boxes indicate data points between the first and third quartiles of the distribution, whiskers show minimum and maximum values, black horizontal lines and '+' sign indicate the median and mean values respectively. Data in **b**, **c** and **e** were analyzed using Unpaired Student's t-test from the total number of spermatocytes (shown as 'n') from multiple animals indicated as 'N'.

aberrations, we immunolabeled pachytene spermatocyte chromosome spreads to quantify MLH1 foci, a proxy for "Type 1" crossovers (COs), which are subject to interference and constitute ~90% of all CO[40]. In the two most severe mutants *Mlh3*[R1230H/R1230H] and *Mlh1*[K618T/K618T], chromosomes exhibited normal homolog pairing and synapsis as reported for knockouts[11,14,21], but had markedly reduced MLH1 foci (Fig. 2a, b; 2.9 and 7.7 per cell, respectively, vs. 21.4 foci in WT). Such a severe reduction in 'obligate' class I COs might either trigger apoptosis at metaphase I, or prevent zygotic progression after fertilization[21,26].

The *Mlh1*[V716M] and *Mlh1*[E268G] mutants, though less affected, also had significantly fewer total MLH1 foci per spermatocyte on average. Since even small reductions in CO recombination might lead to some chromosomes lacking an obligate CO (chiasma), we quantified MLH1 focus-deficient chromosomes. Five of the seven mutants examined exhibited a significant increase in MLH1 focus-deficient chromosomes (Fig. 2c). This suggests an increase in spermatocytes bearing one or more achiasmate chromosomes. Consistent with this hypothesis, *Mlh1*[K618T/K618T] spermatocytes exhibited an average of ~2 fewer metaphase I chiasmata than

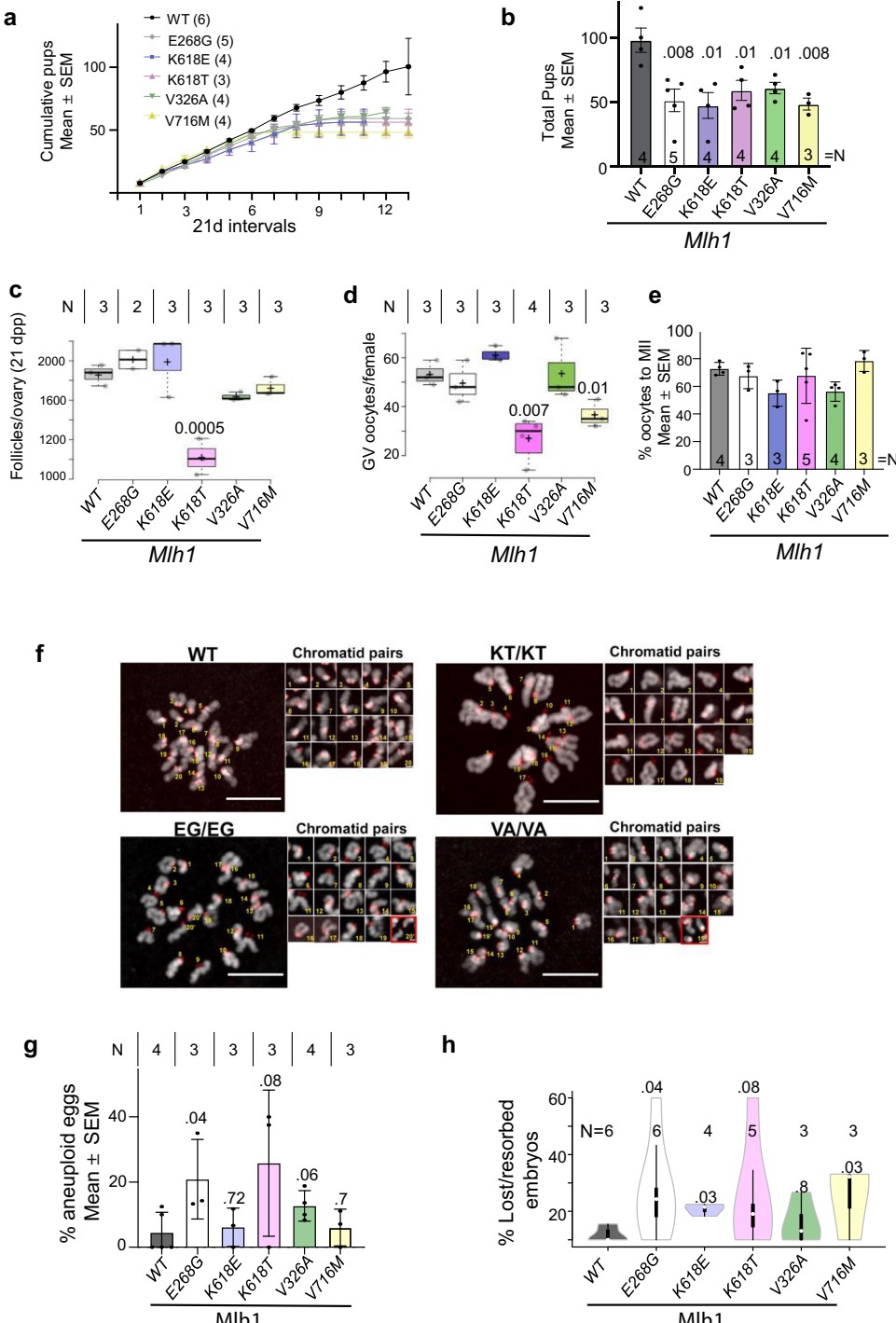

controls, leading to a significant decrease in bivalent chromosomes (increase in univalents; Fig. 2d, e).

**Age-dependent decline in reproductive capacity of *Mlh1/3* mutant females associated with increased oocyte aneuploidy.** In *Mlh1* or *Mlh3* knockouts, females are sterile because their oocytes cannot form a normal spindle and chromosomes do not properly align to the meiotic metaphase I plate[30,41]. This near complete absence of bi-oriented, CO-tethered bivalents at diakinesis can activate quality control mechanisms, namely the Spindle Assembly Checkpoint (SAC), causing meiotic arrest[26,28]. The SAC appears to be more stringent in spermatocytes than oocytes[27,29,42,43], thus eliminating defective spermatocytes

whereas oocytes attempt to complete gametogenesis. To test potential impacts on fertility of females bearing the humanized *Mlh1/3* alleles, we evaluated fecundity of the $Mlh1^{E268G}$, $Mlh1^{V326A}$, $Mlh1^{K618E}$, $Mlh1^{K618T}$, and $Mlh1^{V716M}$ mutants. Interestingly, litter sizes of all mutant females declined over time compared to controls (Fig. 3a) and overall, they had only ~ half of the total offspring as controls over the 36 weeks they were housed with WT sires (Fig. 3b). The overall shortfall became dramatic when mutant dams were 5–6 months old, and this shortfall was largely attributable to the following: (i) decrease in fecundity after the first ~six litters (Fig. 3a); (ii) longer intervals between litters; and (iii) earlier cessation of litters, despite evidence for continued ovulation (Supplementary Fig. 4). Collectively, these resulted into

**Fig. 3 Age-dependent subfertility in *Mlh1* mutant females and increased aneuploidy in oocytes. a** Litter sizes of mutant females over their reproductive lifespans. Eight-week old wild-type and littermate mutant females were mated with wild-type males, and the number of pups born per litter are plotted. For all panels, the following abbreviations apply to genotypes: E268G, $Mlh3^{E268G/E268G}$; K618E, $Mlh1^{K618E/K618E}$; K618T, $Mlh1^{K618T/K618T}$; V326A, $Mlh1^{V326A/V326A}$; V716M, $Mlh1^{V716M/V716M}$. Numbers of animals examined for each genotype are shown in parentheses. **b** Total pups delivered by females during their reproductive lifespan (~2 to 10 months, see Methods). **c** Follicle counts (all stages; the great majority were primordial follicles) in three-week old females. *p*-values < 0.05, in comparisons of mutant to WT, are indicated. Dpp, days post-partum. N indicates animals/group. One ovary from each animal was prepared for histological quantification. **d** Number of total germinal vesicle (GV) stage oocytes collected from superovulated WT and mutant females (3–4 month old). *p*-values < 0.05 relative to WT (+/+) are shown. Horizontal lines are mean and whiskers are ± Std. dev. **e** Percent of GV oocytes (shown in **d**) that matured to metaphase meiosis II (Met II). *p*-values < 0.05 relative to WT (+/+) are shown. **f** In situ metaphase chromosome spreads showing abnormalities in mutants. Oocytes were stained to detect centromeres (ACA, red) and DNA (DAPI, gray). Representative confocal z projections are shown. Sister chromatid pairs are numbered and insets represent each sister chromatid pair in MII. The red squares indicate prematurely separated sister chromatid pairs. Scale Bar = 10 μm in main panels, and 2 μm in insets; **g** Percentage of aneuploid metaphase II oocytes from (E). *p*-values are shown above bars. **h** Violin plots of embryo loss estimates in *Mlh1* mutants. Females homozygous for the indicated mutations (3–4 mth old) were mated to C57BL/6 J (WT) males, sacrificed at E13.5, and the number of viable embryos scored, as well as ovulation sites on the ovary. The difference between the two were presumed to have died during gestation at various stages and are plotted on the Y axis. The control females ("WT") were +/+ or heterozygous littermates. *p*-values are shown and were calculated from the Student's two-tailed *t*-test. Box limits in **c** and **d** indicate the 25th and 75th percentiles. Whiskers in **h** extend 1.5 times the interquartile range of the 25th and 75th percentiles; polygons in h represent density estimates of data and extend to extreme values. N values are number of pregnancies examined. Data in **b**–**e**, **g** and **h** were analyzed using Unpaired Student's *t*-test from multiple biological replicates shown as 'N'.

1.6–2-fold fewer pups from all 5 mutant dams compared to WT (Fig. 3b). The most severe phenotypes were displayed by $Mlh1^{K618T}$ and $Mlh1^{V716M}$ dams. After their first five litters, these dams produced 2.2- fold and 6.5-fold fewer pups than WT, respectively, (22 ± 6.1; 7.7 ± 2.6; 49.8 ± 8 SEM).

Whereas homozygotes of the aforementioned five alleles showed age-related decrease in litter sizes, only $Mlh1^{K618T/K618T}$ females had dramatically fewer (~half) follicles (combined primordial and developing follicles, Fig. 3c) at three weeks of age. While we did not perform longitudinal studies on the $Mlh1^{K618A}$ allele, female homozygotes were fertile. In preliminary crosses, two young homozygous females had apparently normal litter sizes—an average of 7.5 pups/litter ($N = 5$ litters)—when crossed to heterozygous males.

It is possible that in the $Mlh1^{K618T}$ mutants with reduced follicles at three weeks of age (and which also had the greatest reduction in COs in males), oocyte loss was triggered by failure to repair CO-designated DSBs, and subsequent activation of the meiotic DNA damage checkpoint[44]. Alternatively, the smaller litter sizes from mutant dams might be attributable decreased COs and thus increased aneuploidy in ovulated eggs with age, although it is unclear how this would occur. An increase in aneuploidy could lead to increased embryonic lethality as with *Sycp3* null females[9].

To test the hypothesis that reduced litter sizes are attributable to aneuploidy-induced embryo loss, we first examined the fraction of aneuploid eggs produced by mutant females (3–4 months of age). Following hormone stimulation, oocytes were matured in vitro to MII, then spread to visualize chromosomes. All mutants examined except $Mlh1^{K618T}$ and $Mlh1^{V716M}$, both of which had smaller ovarian reserves (see above), contained similar numbers of prophase I arrested oocytes as WT females (Fig. 3d). About 70% of the oocytes from all genotypes matured to MII after 15 h (Fig. 3e), although maturation, as marked by polar body extrusion, was ~20% less efficient for the $Mlh1^{K618E}$ and $Mlh1^{V326A}$ genotypes. Next, we evaluated the ability of these females to produce euploid eggs (20 pairs of sister chromatids), and then we quantified the number of sister chromatid pairs in eggs from each mutant. The $Mlh1^{E268G}$, $Mlh1^{K618T}$ and $Mlh1^{V326A}$ alleles presented marked increases in aneuploid eggs, detected as abnormal number of pairs of sister chromatids (Fig. 3f and insets, g; 20.9 ± 7.1%, 25.8 ± 12.9% and 12.7 ± 2.3, respectively, compared to 4.5 ± 2.7% in WT). Moreover, eggs from $Mlh1^{E268G}$ and $Mlh1^{V326A}$ alleles presented an

**Table 3 Embryo loss during development.**

| Genotype | Pregs | CL (Avg) | Embryos (Avg) | % embryo loss |
|---|---|---|---|---|
| WT | 6 | 59 (9.8) | 57 (9.5) | 3.4 |
| $Mlh1^{E268G}$ | 6 | 58 (9.7) | 38 (6.3) | 34.5 |
| $Mlh1^{K618E}$ | 3 | 37 (12.3) | 29 (9.7) | 21.6 |
| $Mlh1^{K618T}$ | 5 | 46 (9.2) | 39 (7.8) | 15.2 |
| $Mlh1^{V326A}$ | 4 | 35 (8.8) | 31 (7.8) | 11.4 |
| $Mlh1^{V716M}$ | 3 | 33 (11) | 23 (7.7) | 30.3 |

All mutant genotypes are homozygous. *Pregs* pregnancies dissected, *CL* Corpora lutea, *Avg* average. The embryos counted were viable when dissected at E13.5. The % embryo loss = (#CL-#embryos)/#CL * 100.

abnormal phenotype of prematurely separated sister chromatids (Fig. 3f insets, outlined red).

We next explored whether the high levels of aneuploidy in eggs from mutant females led to increased embryonic death of embryos. Control and mutant females that were 3–4 months old (similar to the age of females used in the ploidy experiments) were mated with wild-type males, dissected at E13.5, and the numbers of viable embryos and corpora lutea (CL; reflecting the number of ovulated oocytes) were counted. The fraction of embryonic lethality was inferred as: [CL-embryos]/CL[45]. Whereas lethality was only 3.4% in WT mothers, it was several fold higher in the mutants, ranging from 11.4% in $Mlh1^{V326A}$ to 34.5% in $Mlh1^{E268G}$ (Table 3). These results suggest that mutations in *MLH1/3* can elevate the incidence of aneuploid oocytes, leading to a corresponding increase in spontaneous pregnancy loss.

## Discussion

The process of meiotic recombination, specifically crossing over, is essential for fertility and the production of viable, healthy offspring. A severe reduction in COs will prevent proper alignment of meiotic chromosomes at the metaphase I plate, leading to arrest or failed fertilization. However, if only one or a very few chromosomes lack a chiasma, viable gametes can still form but will be prone to mis-segregation of parental homologs; this can lead to aneuploid embryos. A recent study of recombination in human fetal oocytes revealed that having 10–20% fewer COs (MLH1 foci) overall was associated with such oocytes having 1–2 chromosomes completely lacking a CO[46]. Given that the most

common cause of miscarriage (~60%) is embryonic aneuploidy, primarily trisomies[1], it is of interest whether the frequency of so-called "exchangeless" chromosomes is genetically controlled. There are indications that parental genetic factors can predispose to generation of aneuploid conceptuses, and may contribute to cases of RPL[5,6]. In this respect, variants impacting genes important for recombination, including the MUTL homologs, are potential candidates for influencing aneuploidy rates.

SNPs in MLH1 and MLH3 have been statistically associated with male infertility in some studies focusing on candidate genes (i.e., not GWAS), but there have not been validation studies[47–49]. For female fertility, GWAS for phenotypes such as age to menopause, polycystic ovary syndrome, and endometriosis have been performed, but only a few associations have been reported for genes involved in oogenesis per se[50]. In this study, we focused on human Variants of Unknown Significance (VUS) in MLH1/3, with the goal of determining possible effects on fertility. Because our earlier studies of VUS in essential fertility genes indicated that functional prediction algorithms alone overpredicted deleteriousness[33], we applied additional functional criteria to select variants for mouse modeling, namely, studies of analogous mutations in yeast or disruption of protein-protein interactions in Y2H assays. Given that most of the alleles reported here that exhibited defects in these assays caused phenotypes in mice, we believe that this approach is valuable for prioritizing experimentation.

In our opinion, the most intriguing result from this study is that female fecundity in several mutants declined with age and reproduction ceased prematurely (Fig. 3a, b), despite having no apparent shortfall (compared to WT) in the ovarian reserve at wean age (Fig. 3c). One possibility is that after wean age (~3 weeks), oocyte attrition accelerated in the mutants, such that they exhausted their ovarian reserve. However, histology of these older females revealed numerous corpora lutea, indicative of continued ovulation in these aged females (Supplementary Fig. 5). Future experiments aimed at quantifying the reserve, numbers of ovulated eggs, the ability of these eggs to become productively fertilized, and other markers of ovarian aging could reveal the basis of this phenomenon.

An alternative explanation is that the quality of oocytes recruited later in life might be lower than at earlier stages in the mutants. We can only speculate as to the nature of such quality differences, but they might be related to increased aneuploidy caused by prolonged periods in primordial follicles with achiasmate chromosomes. It is also possible that these mice might represent an example of the "production line" hypothesis that was originally proposed as an explanation for increased aneuploidy in women as they age[51]. This hypothesis posits that eggs are ovulated in the order that they originally proceed through meiosis in utero, and that the "later" oocytes suffer from deficient recombination, and thus higher aneuploidy at MI. While this hypothesis has been refuted in humans[52], there is evidence that in mice, the first oogonia to enter meiosis are the first to be ovulated[53]. Nevertheless, there is no evidence relating crossover rates to time of meiotic entry. Interestingly, a similar phenomenon was observed in Sycp3 null mice, where an age-correlated decrease in litter sizes was attributable to increased embryonic death from aneuploidy[9]. Additionally, a small scale study implicated mutations in SYCP3 as being responsible for RPL in two women[7]. It has been hypothesized that women with smaller ovarian reserves have an increased chance of having aneuploid abortuses[54]. In future work, our models may be useful in exploring this possible relationship in aging mice, and to better understand sexually dimorphic consequences upon gamete quality and checkpoint mechanisms.

Inherited mutations in MLH1 can cause Lynch Syndrome, sometimes referred to as hereditary non-polyposis colorectal cancer (HNPCC), although patients are also subject to a variety of cancers including endometrial[55]. Lynch Syndrome exhibits a dominant mode of inheritance, and the tumors are characterized by microsatellite instability (MSI). Tumor initiation is attributable to either epigenetic silencing or mutation of the normal MLH1 allele inherited from one parent[55–57]. Mice null for Mlh1 exhibit MSI, increased morbidity (moribund at an average age of 7 months), and develop lymphoma and various types of tumors including gastrointestinal[58]. However, we did not observe an increased incidence of tumors in any lines or microsatellite instability in the mutants tested (data not shown). Mutations in MLH3 are not associated with Lynch Syndrome, but there is one report that null mice are predisposed (50%) to gastrointestinal tumors with a long latency[59]. Although the orthologous yeast mlh3 alleles have MMR defects (Table 1), we did not observe tumors or premature morbidity in our Mlh3 mouse mutants. However, we did not age them beyond 1 year. It is also possible that for both sets of models (Mlh1/3), that the genetic background we used was not as tolerant to transformation (at least before 1 year of age) as those in published studies. Because of these caveats, we cannot conclude that these alleles do not cause Lynch Syndrome in humans.

As mentioned in the Results, when this project began, we selected the two variants at K618 (K618E and K618T) that remain present in dbSNP as rs35001569 and rs63750449, respectively. However, after the mouse models were created, the gnomAD database was revised to indicate that these two variants co-occur (are in phase), encoding a K618A amino acid change at essentially identical frequencies in populations (Table 1). The question remains as to whether the $MLH1^{K618E}$ and $MLH1^{K618T}$ exist at all in populations, or at very low frequencies (approximately 100-fold lower based on the numbers cited above). Notably, the $MLH1^{K618E}$ and $MLH1^{K618T}$ alleles are listed in the ClinVar database, and reported in patient samples[60–62]. It is unclear if the three alleles arose independently, or if they were derived from one another. Regarding the $MLH1^{K618A}$ allele and its relationship to Lynch Syndrome, which is somewhat controversial, a recent large study found no evidence for this allele's involvement, and thus concluded that $MLH1^{K618A}$ is benign[63]. Interestingly, a computational and experimental study of MLH1 variants suggested that many variants that cause Lynch Syndrome do so by thermodynamically de-stabilizing the MLH1 protein, leading to increased degradation[64]. Abildgaard and colleagues[64] tested four of the alleles modeled here ($MLH1^{E268G, K618T, K618A and V716M}$), and each were classified as being "likely benign" by virtue of the proteins having substantially normal stability.

In sum, these studies indicate that variants in the MUTLγ (MLH1/3) complex genes can have subtle-to-severe effects on gametogenesis that may go unnoticed without rather detailed phenotyping, including aging studies. Homozygosity for most of the alleles did not cause infertility or drastic reductions in ovarian reserve or sperm counts, although it is conceivable that some of these alleles in trans to a null allele, or in the context of a variant in the other member of the MUTLγ complex, could cause synthetic phenotypes. Nevertheless, the phenotypes of increased aneuploidy in oocytes, and reduced reproductive lifespan of females are highly relevant to the human condition, and thus these mouse models can be valuable for assessing risk at an early age in women, and to dissect the biology of these phenomena.

## Methods
**Construction of baker's yeast strains to measure meiotic crossing over and DNA mismatch repair.** The SK1 background baker's yeast strain EAY3255 was constructed for the analysis of MMR and meiotic crossing over phenotypes[65]. These and all yeast strains generated/used are listed in Supplementary Table 2.

EAY3255 contains a spore autonomous fluorescent protein (Tomato) marker located near the centromere on chromosome VIII[38] and a *lys2::InsE-A14* cassette to measure reversion to Lys+ [37]. *mlh3* alleles cloned into pEAI plasmids (*mlh3-X::KANMX*) were transformed into EAY3255 by digesting each plasmid with *Bam*HI and *Sal*I to generate an *mlh3-X::KANMX* fragment. Approximately 1 µg of each digested plasmid was mixed with 40 µg of salmon sperm carrier DNA (Invitrogen, San Diego, CA, boiled just before addition) and added to ~$10^8$ yeast that had been grown to mid-log in yeast extract-peptone-dextrose (YPD) media. The cell-DNA mixture was suspended in a solution containing 0.1 M lithium acetate, 10 mM Tris 8.0, 1 mM EDTA, 40% polyethylene glycol[66]. The resulting mixture was incubated for 40 min at 42 °C then allowed to recover overnight in 5 ml of YPD media and then plated onto a plate containing geneticin (G418; Invitrogen) at 200 µg/ml. At least two independent transformants for each genotype (verified by DNA sequencing) were made.

EAY3255 and derivatives were used to measure the effect of *mlh3* mutations on *lys2::InsE-A14* reversion rate (see below). To measure meiotic crossing over, strains from the EAY3255 background were each mated to EAY3486, an SK1 *mlh3Δ* strain containing the m-Cerulean fluorescence marker located at *THR1* on chromosome VIII, ~20 cM away from the Tomato marker present at *CEN8* in EAY3255. Diploids were selected on yeast synthetic minimal media lacking tryptophan and leucine and maintained as stable strains. Diploids were then patched onto YPD media for growth for two days at 30 °C. Meiosis was induced upon patching the diploid strains from YPD onto sporulation media (1% potassium acetate; 0.1% yeast extract; 0.05% glucose; 0.021% complete amino acids (2% Bacto Agar (Difco)) and incubated at 30 °C for 48–72 h prior to analysis. Wild-type strains carrying the fluorescent protein markers used to make the above test strains were obtained from Scott Keeney[38].

**Spore autonomous fluorescent protein expression assay to measure meiotic crossing over.** Tetrads obtained from EAY3255/EAY3486 background diploids were treated with 0.5% NP40 and briefly sonicated before analysis by fluorescence microscopy. Tetrads were analyzed using a Zeiss AxioImager M2 microscope equipped with RFP and CFP filters. At least 250 tetrads for each *mlh3* allele were counted to determine the % tetratype (two spores show the parental genotype and two show a configuration consistent with a crossover event between the Tomato and m-Cerulean markers located on Chromosome VIII; Supplementary Table 1). Two independent transformants were measured per allele, measured on at least two days with a similar amount analyzed to avoid batch effects. A statistically significant difference (p<0.002) from wild-type and *mlh3Δ* controls, based on $\chi^2$ analysis (Pearson $\chi^2$ contingency test, with a Bonferonni correction for 26 comparisons), was used to classify each allele as exhibiting a wild-type (+), intermediate (+/−, +/−), or null (-) phenotype.

***lys2A14* reversion assay to monitor MMR.** The EAY3255 derived haploid strains described above were analyzed for reversion to Lys+. Briefly, strains described in Supplementary Table 2 were freshly struck from frozen stocks and grown in complete synthetic minimal media to ~2 mm in diameter colonies (~3 days growth at 30 °C) and then individual colonies were inoculated in 3 ml of yeast synthetic minimal media and grown overnight at 30 °C (~16 h). Overnight cultures were plated with appropriate dilutions onto plates with complete synthetic minimal media and synthetic minimal media lacking lysine to measure *lys2::insE-A14* reversion. Colonies were counted after 3 days of incubation at 30 °C. Rates of *lys2::insE-A14* reversion were calculated as µ=f/ln(N·µ), where f is reversion frequency and N is the total number of revertants in the culture[37]. 11 to 16 independent cultures were analyzed for each mutant allele as well as wild-type and *mlh3Δ* controls. Two independently constructed transformants for each allele were analyzed on at least two days to avoid batch effects, with a similar number of repetitions performed each day. Reversion rates (Supplementary Table 1) were measured and the median rate for each genotype was normalized to the wild-type median rate (1X) to calculate fold increase. Alleles were classified as wild-type (+), intermediate (+/−), or null (-) based on the 95% confidence intervals (CI)[67]. If the 95% CI for the allele overlapped with the 95% CI for one of the controls (wild-type or null) they were considered statistically equivalent to the respective control.

**Variant selection for *MLH1*.** To select *MLH1* SNPs for cloning and subsequent Y2H testing, we filtered gnomAD v2.1 for missense mutations with overall AF between 0.02% and 2%. 15 missense SNPs fell within this range, of which 13 were successfully cloned and tested by Y2H.

**Generating mutant *MLH1* clones.** Clones for *MLH1* and all Y2H-tested interaction partners were obtained from hORFeome v8.1[68] or v5.1[69]. The *MLH1* alleles were generated by site-directed mutagenesis. Primers for mutagenesis were designed using the webtool primer.yulab.org. To minimize sequencing artifacts, PCR was limited to 18 cycles using Phusion polymerase (New England Biolabs, M0530). Mutagenesis PCR product was then digested overnight using *Dpn*I (New England Biolabs, R0176) followed by bacterial transformation into competent cells. Single colonies were recovered on LB agar plates containing spectinomycin. Four colonies were picked per mutagenesis attempt and then were validated to contain

the mutation of interest through Sanger sequencing using primers designed to cover the entire *MLH1* sequence.

**Profiling disrupted protein interactions through Y2H.** Wild-type and successfully mutated *MLH1* clones were transferred into Y2H vectors pDEST-AD and pDEST-DB by Gateway LR reactions then transformed into *MAT*a Y8800 and *MAT*α Y8930, respectively. Testable Y2H interaction partners were identified by surveying a Y2H reference interactome comprised of interactions reported in four publications[70–73]. All ORFs corresponding to interactions with MLH1 were then selected for Y2H testing. MLH1 interaction partners, transformed into *MAT*a Y8800 and *MAT*α Y8930 were then mated against AD- and DB-MLH1 transformed yeast of the opposite mating type in a pairwise orientation by co-spotting yeast inoculants onto YEPD agar plates. To screen for autoactivators, DB-MLH1 yeast cultures were also mated against *MAT*a yeast transformed with empty pDEST-AD vector. Mated yeast were incubated overnight at 30 °C and then replica-plated onto dropout Synthetic Complete agar media lacking leucine and tryptophan (SC-Leu-Trp) to select diploid yeast. After overnight incubation at 30 °C, diploid yeast were then replicate-plated onto dropout SC agar media lacking leucine, tryptophan, histidine, and supplemented with 1 mM of 3-amino-1,2,4-triazole (SC-Leu-Trp-His+3AT). Diploid yeast were also replica-plated onto SC agar media lacking leucine, tryptophan, and adenine (SC-Leu-Trp-Ade). After overnight incubation at 30 °C, plates were replica-cleaned and incubated again for three days at 30 °C. An interaction was scored as disruptive only if (1) mutant MLH1 reduces growth by at least 50% relative to its corresponding wild-type interaction, and (2) neither wild-type nor mutant DB-MLH1 scored as an autoactivator.

**Production of 'humanized' mouse-lines and mouse breeding.** All 'humanized' alleles were generated by CRISPR/Cas9-mediated genome editing[33,74]. In particular, high-quality gRNA seed sequences were selected using the Benchling (https://benchling.com) guide design tool, and sgRNAs were produced via cloning-free in vitro transcription[75]. Cas9 protein, sgRNA and single stranded oligodeoxynucleotides (ssODN) bearing the desired sequence changes (oligos listed in Supplementary Table 3) were microinjected into single cell zygotes produced from matings between FVB/NJ and B6(Cg)-Tyr^c-2J/J inbred mice. Correctly edited founder (F0) animals were backcrossed to B6(Cg)-Tyr^c-2J/J for two generations, then maintained in a mixed strain background (C57BL/6 J x FVB/NJ).

**Fertility tests and embryo loss.** Control and experimental animals (8–10 weeks old) were housed with age-matched fertile mates (C57BL/6J) for up to 10–12 months of age. Litter sizes were determined by counting pups on the day of birth.

For embryo loss analyses, homozygous females (12–16 weeks old) were mated to WT males and the presence of a copulation plug in the morning was recorded as 0.5 dpc. Females were sacrificed at 13.5 dpc for counting of both implantation sites and corpora lutea (CL; indicating the numbers of ovulated oocytes). Graphs and statistical analyses were performed with GraphPad Prism5.

**Ovary histology and follicle quantification.** Ovaries were collected from 3-week females and were fixed in Bouin's for 24 h, washed in 70% ethanol, paraffin embedded, and further serial sectioned at 6 µm followed by staining with hematoxylin and eosin (H&E). Every fourth section was counted under light microscopy. Though we reported the combined number of follicles in Fig. 3, we maintained records of the follicle subtypes (e.g., primordial, primary, secondary, antral) and can provide them upon request.

**Surface spread preparation and immunocytology.** To prepare prophase I surface spreads[76], mouse seminiferous tubules were incubated in a buffer consisting of 30 mM Tris pH7.2, 50 mM sucrose, 17 mM trisodium dehydrate, 5 mM EDTA, 0.5 mM DTT, 0.1 mM PMSF, pH8.2-8.4) at room temperature for 1 h. Small sections of testis tubule were dissected in 100 mM sucrose and spread onto slides coated with 1% paraformaldehyde and 0.15% Triton X, then incubated in a humid chamber for 2 h at room temperature. For all experiments, at least 3 males from each genotype were evaluated. Following final washing of chromosome slides in 0.4% Photo-Flo 200 (Kodak 1464510) for 2 × 5 min, slides were air dried for ~10 min and stored in −80 °C or used immediately for staining.

For staining, slides were washed in 1xPBS, 0.4% Photoflo for 10 min, followed by 2 × 10 min. washes in 1x PBS, 0.1% Triton X and 1 hr incubation in blocking buffer (3% BSA, 10% Goat Serum, 0.0125% Triton X, 1 x PBS), at 37 °C. Finally, all antibodies were diluted in blocking buffer, and slides were incubated for 12 hr at RT.

Primary antibodies used were: rabbit anti-SYCP3 (1:500, ab15093; Abcam), mouse anti-SYCP3 (1:500, ab97672; Abcam), and mouse anti-MLH1 (1:50, 550838; BD Pharmingen). Slides were washed and incubated for 1 hr with the following secondary antibodies: goat anti rabbit-IgG 488 (1:2,000, A11008; Molecular Probes), goat ant-rabbit IgG 594 (1:1,000, A11012; Molecular Probes), and goat anti-mouse IgG 594 (1:1,000, A11005; Molecular Probes). Nuclei were counterstained with DAPI, and slides were mounted with ProLong Gold antifade reagent (P36930; Molecular Probes).

**Sperm counting**. Cauda epididymides were dissected and minced into 2 mL of PBS. Sperm were allowed to swim into the buffer via 15 min incubation at 37 °C. The sperm were diluted PBS, incubated for 1 min at 60 °C, and then counted with a hematocytometer.

**Testes histology and Immunohistochemistry**. For histological analyses, testes were fixed for 24 h at room temperature (RT) in Bouin's solution. Tissue was further paraffin-embedded, sectioned (7 μm), and stained with H&E. For immunohistology, testes were essentially fixed in 4% paraformaldehyde for ~24 h. Paraffin-embedded tissues were further sectioned at 7 μm. For TUNEL and PH3 double immunostaining, sections were deparaffinized followed by antigen retrieval using Sodium Citrate Buffer (10 mM Sodium Citrate, 0.05% Tween 20, pH 6.0). Sections were blocked in PBS containing 5% goat serum for 1 h at RT, followed by TUNEL staining, performed following instructions manufacturer's instruction (Invitrogen, Click-iT™ TUNEL kit). This was followed by 1° antibody incubation overnight at 4 °C (anti-pH3 (Ser10), 1:100, Millipore) and anti-rabbit AF488 (Invitrogen, 1:1000) 2° antibody incubation for 1 h, RT. Slides were counterstained using DAPI, mounted in Vectashield and imaged as described in below.

**Imaging and analysis of testes cross sections**. Testes were fixed in Bouin's, embedded in paraffin and stained with hematoxylin and eosin (H&E) in Cornell's Diagnostic Pathology Lab. For TUNEL labeling, testes cross sections were double immunolabeled with TUNEL and anti-pH3 (see above) antibody and were imaged using a confocal microscope (u880, Carl Zeiss, Germany) with a Plan Apo 40× water immersion objective (1.1 NA) and Zen black software. The following lasers were used: argon laser-488 nm, blue-diode-405 nm, DPSS laser—561 nm. Following identical background adjustments for all images, cropping, color, and contrast adjustments were made with Adobe Photoshop CC 2017.

**Oocyte in vitro maturation and in situ chromosome counting**. Prophase I-arrested oocytes (GV) were collected following standard methods[77] as follows. Forty-eight hours prior to collection, females were injected intraperitoneally with CARD HyperOva (KYD-010-EX-X5; CosmoBio USA) consisting of Inhibin antiserum and equine chorionic gonadotrophin (Sigma, 230734). Further, ovaries from 12–16 weeks old females were dissected and GV oocytes were released from the follicles by puncturing the ovaries several times with 27-gauge sewing needles in MEM/polyvinylpyrrolidone media containing 2.5 μM milrinone (Sigma-Aldrich; M4659) to prevent oocyte maturation after the release from cumulus cells. Then, cumulus cells were removed by gentle pipetting and fully-grown oocytes were matured in vitro in Chatot, Ziomek, and Bavister (CZB) media without milrinone in a humidified incubator programmed to 5% $CO_2$ and 37 °C for 14 h until they arrived to metaphase II. They were cultured for 2 h in 100 μM Monastrol (Sigma, M8515) to disorganize the spindle and separate the chromosomes. Then, eggs were fixed in 2% paraformaldehyde in PBS for 20 min and permeabilized in PBS containing 0.2% Triton X-100 (Sigma-Aldrich, 900-93-1) for 20 min. Eggs were stained with the anticentromere antibody (ACA) to detect centromeres (Antibodies Incorporated; #15-234; 1:30) and DAPI to detect DNA. Chromosome numbers were quantified in metaphase II eggs[78]. Mouse eggs should have 20 pairs of sister chromatids; any deviation of this number was considered aneuploid. Eggs were imaged with the i880 (Zeiss) confocal at 0.5 μm z-intervals. Chromosome counting was performed with NIH Image J software, using cell counter plugins.

**Animals**. All animal strains were maintained in a mixed strain background (C57BL/6 J x FVB/NJ). We have complied with all relevant ethical regulations for animal experiments, and these experiments were performed under a protocol (2004-0038) approved by Cornell's Institutional Animal Care and Use Committee (IACUC), which ensures ethical animal research use. Mice were housed at 23 °C with a light cycle of 10 h dark and 14 h light.

**Statistics and reproducibility**. All data presented in this manuscript is generated from at least 3 or more biological replicates, unless otherwise noted. Statistics is performed on datapoints with ≥3 biological replicates. Each mouse is considered a biological replicate.

**Reporting summary**. Further information on research design is available in the Nature Research Reporting Summary linked to this article.

## Data availability
All relevant data such as litter sizes and sperm numbers are present in the manuscript. Fluorescent images (immunolabelings) from which numbers were generated (such as MLH1 foci) are stored by the authors and primary images can be provided upon request. Source data are included with this manuscript. All unique materials used are readily available from the authors, with the exception of mouse strains, which may require extended periods of time for reanimation and/or breeding before providing to requestors. Source data are provided with this paper.

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

## Acknowledgements

We thank S. Keeney for yeast strains, R. Munroe and C. Abratte in Cornell's transgenic core for making mice, and P. Cohen for critical reading of the manuscript in advance of submission. This work was supported by grants R01HD082568 to J.S. and H.Y., R01HD091331 to K.S., and R35 GM134872 to E.A.

## Author contributions

P.S. characterized all *Mlh1* and the *Mlh3^A1394T* mouse mutants and contributed to writing the manuscript. R.F. performed the Y2H experiments and wrote relevant sections of the manuscript; H.Y. supervised R.F. and was involved in SNP selection metrics; C.S.B. (with P.S.) performed the oocyte development and ploidy experiments presented in Fig. 3d–g, was supervised by K.Schindler, and C.S.B. and K.S. wrote relevant parts of the manuscript. T.N.T. analyzed *Mlh3* mouse mutants not done by P.S. G.P. and N.A-S. performed the yeast studies of mismatch repair and recombination, and were supervised by E.A.A., who wrote relevant parts of the manuscript and design/interpretation of the yeast experiments. K.J.S. assisted with mouse husbandry. J.C.S. oversaw the overall project, its design, and had a major role in writing and revising the manuscript.

## Competing interests

The authors declare no competing interests.

**Additional information**

