## [Peer Review File · Nature Communications]

REVIEWER COMMENTS

Reviewer #1 (Remarks to the Author):

Mutations in MLH or MSH genes results in defects in postreplicative mismatch repair and/or meiotic recombination. MLH1 and MLH3 factors analyzed in this study have function in both processes. Correspondingly, previous mouse knockout models identified cancer predisposition and fertility issues. Several rather preliminary studies also linked defects in these proteins with reduced fertility in humans.

Here, the authors selected numerous variants of unknown significance, which were extensively profiled based on homology for meiotic and mismatch repair defects, as well as protein-protein interactions in yeast. The analysis, which is shown in supplementary material, identified several variants with notable meiotic and mismatch repair defects. The selected prospective mutants were introduced into mice. The manuscript then describes the effects of the selected variants on reproductive parameters in both males and females. Although the extent of defects differed among the variants, multiple mutants exhibited significant abnormalities. Unexpectedly, it was noted that several variants exhibited maternal age-dependent defects, showing that MLH1 and MLH3 relate to the effect of maternal age, although the mechanism remains undefined.

Overall, the study is designed well, the experiments are of a high quality. This study convincingly links human SNPs in MLH1 and MLH3 genes with reproductive defects in mammals. That said, a negative point is that the results were expected: knockouts of these genes are well-known to cause infertility in mice, so mutations affecting conserved residues affecting key functions of the proteins (nuclease, ATPase activities) are expected to also have notable phenotypes.

- If space permits, some of the yeast data might be moved to main data figures.

- The authors should compare (at least by referring to literature) the extent of defects of the analyzed alleles with null (knockout) phenotypes.

- Mutation of K618: The K618T variant gives the strongest phenotype of all MLH1 mutants and stands out in most assays (Fig. 1-3). Yet, it is noted that this variant does not exist in the population, and it is rather present as K618A. Surprisingly, my understanding is that K618A variant does not present abnormal phenotype. How can that be? Is it possible that the most serious mutations cause embryonic lethality in humans, and only "weaker" mutations then associate with infertility?

Reviewer #2 (Remarks to the Author):

The funding function of meiosis is to keep homologs joined to allow chromosome reduction. Genetic variation at the essential miss match repair genes MLH1 and MLH3 might potentially be responsible of human infertility and / or RPL and miscarriages. Based on this hypothesis, the authors evaluated human missense SNPs curated in gnomAD databases. By in silico modelling and biochemical assays / yeast genetic assays the authors selected those more severe variants for modeling in mice. The results show that the mouse models exhibited crossing over defects that led to age-related decreases in female fecundity, elevated aneuploidy, and consequent increases in pregnancy loss. The work is very well designed and the analysis is well conducted. The results are of great relevance in the field of human biology and reproduction and demonstrate that human genetic variants (relatively common) in the meiotic recombination machinery can lead to human infertility and miscarriages. Interestingly, these genetic variants in females led to age-dependent subfertility and aneuploidy. This observation is novel and of great relevance and provide a novel hypothesis to explain age-dependent RPL and trisomies such as Down syndrome. The MS is thus of great interest and relevance for the scientific community.

I have some few minor comments:

-It would be of help for understanding the Y2H disruption section to incorporate in the Result section a brief explanation of how / why those genes/proteins were selected for this analysis (beyond mat and methods).

-From my perspective, I miss a rationale explanation for the different pipeline used for the selection of the MLH1 and the MLH3 variants. This explanation would clarify why this has been done in such a way.

- The images of the chromosome analysis carried out (figure 3F) to determine the abnormal number of chromatids at MII are not of good quality in my electronic version of the MS. From this figure it is difficult to figure out what is happening at those plates. It should be needed a clearer demonstrative figure.

-Interestingly, despite the absence of reproduction defects in the 6 male mice mutants, histological analysis revealed the presence of abnormal metaphases. In this respect, the confocal analysis carried out does not allow to show in detailed the chromosome segregation defects (S3). Immuno Fluorescence analysis of metaphase I from squashed tubules using for instance SYCP3 and DAPI would notably increase the detailed and resolution of the analysis (are XY more frequently involved?).

-Female analysis for fertility would be improved if it would follow the same script structure used in the male analysis such as the MLH1 staining of COSs. This analysis (CO counts in oocytes) would enable to demonstrate (similarly as it has been shown in males) that COs are reduced or absent and would allow to compare the severity of the phenotypes of each variant in males vs females, which is also of great interest given the known sexual dimorphism of meiotic recombination.

Alberto M Pendas

Reviewer #3 (Remarks to the Author):

In the manuscript titled "MLH1/3 variants causing aneuploidy, pregnancy loss, and premature reproductive aging" by Singh et al., the authors model and study human variants of MLH1/3 in mice. These mismatch repair genes are required for crossovers during meiosis and null alleles cause sterility in mice. The authors perform a rigorous selection procedure involving a combination of computational and molecular assays to choose human population variants of MLH1/3 and then perform a detailed analysis of their functional impact on male and female fertility in mice. They describe the molecular defects caused by the Mlh1 and Mlh3 mutations during male meiosis. In females Mlh1 mutations cause an increase in aneuploidy in oocytes resulting in non-viable embryos. Interestingly, this manifests as an age-related decline in female fertility and the authors discuss the implications of these findings. This is a heroic, unique and thorough study. The manuscript is in good shape and suitable for publication in Nature Communications as is. Few minor comments are listed below.

Minor comments:

1) The schematic for MLH1 variant selection in Figure 1 is hard to follow – please clarify/correct. The text describes selection of variants with AF<2% but the figure shows different section criteria (<1%) for MAF (which I'm assuming is mutation allele frequency). Also the text states that 13 MLH1 variants were tested by Y2H and then those that both disrupted interactions and scored poorly in functional predictions algorithms were modeled, but the pathogenicity algorithms filter is listed at an earlier step in the figure. Also the figure shows 20 SNPs being tested, whereas the text states that 13 SNPs were tested.

2) The text on Page 5 states that MLH3-R1230H mutants show prophase arrest. Do the authors mean metaphase arrest?

3) I think a few sentences describing the histological phenotypes in Figure 1(D) in greater depth would be useful for the reader. At least for the two alleles with strong phenotypes that are described in more detail: MLH3-R1230H and MLH1-K618T

- 4) Figure 2(D) legend: state what the arrows annotate.
- 5) Figure 2(E) is not referenced in the text. And text states that there is a 2-fold reduction in chiasmata in K618T but the figure shows a small median reduction per cell - please clarify.
- 6) The authors state that the Mlh1 mutant females cease to have litters at a younger age compared to wild type despite continued ovulation. A sentence describing the actual data in the text would be nice. How much earlier did the females cease to have litters? Was this phenotype approximately the same for all mutants?
- 7) Figure 3(E) incorrectly references (A), I think the authors meant (D).
- 8) For Figure 3(F, G), it would be useful to have a more detailed description of precisely what staining was scored as aneuploid and a mention of the patterns seen (description of the insets) in the text.
- 9) In Figure 3(H) the whiskers do not seem to represent the maximum and minimum values as stated in the legend.
- 10) I suggest coloring the THR1 locus in Figure S1 the same blue as the blue spores to avoid confusion with purple spores

Response to Reviewers

The authors are grateful for the reviewers taking the time to provide excellent suggestions for improving our manuscript. We are very pleased about the positive perception of the work. Hopefully we have understood the queries and responded satisfactorily.

Reviewer #1 (Remarks to the Author):

Mutations in MLH or MSH genes results in defects in postreplicative mismatch repair and/or meiotic recombination. MLH1 and MLH3 factors analyzed in this study have function in both processes. Correspondingly, previous mouse knockout models identified cancer predisposition and fertility issues. Several rather preliminary studies also linked defects in these proteins with reduced fertility in humans.

Here, the authors selected numerous variants of unknown significance, which were extensively profiled based on homology for meiotic and mismatch repair defects, as well as protein-protein interactions in yeast. The analysis, which is shown in supplementary material, identified several variants with notable meiotic and mismatch repair defects. The selected prospective mutants were introduced into mice. The manuscript then describes the effects of the selected variants on reproductive parameters in both males and females. Although the extent of defects differed among the variants, multiple mutants exhibited significant abnormalities. Unexpectedly, it was noted that several variants exhibited maternal age-dependent defects, showing that MLH1 and MLH3 relate to the effect of maternal age, although the mechanism remains undefined.

Overall, the study is designed well, the experiments are of a high quality. This study convincingly links human SNPs in MLH1 and MLH3 genes with reproductive defects in mammals. That said, a negative point is that the results were expected: knockouts of these genes are well-known to cause infertility in mice, so mutations affecting conserved residues affecting key functions of the proteins (nuclease, ATPase activities) are expected to also have notable phenotypes.

RESPONSE: Thanks for the positive comments. Regarding the expectation that mutations in conserved residues of genes known to be necessary for fertility (like MLH1/3) should have a phenotype: that would seem like a logical assumption, and we indeed also thought so. However, we have been making numerous such alleles in various reproductive genes but have found no deleterious effects (e.g. PMID:26240362, PMID:31393579, PMID:30085085, and another larger set of examples we haven't yet published), underscoring the need for in vivo testing of variants of unknown significance.

- If space permits, some of the yeast data might be moved to main data figures.

RESPONSE: We moved former Table S2 (summary of MMR and HR assays on Mlh3 alleles tested in yeast) to the main text as Table 2.

- The authors should compare (at least by referring to literature) the extent of defects of the analyzed alleles with null (knockout) phenotypes.

RESPONSE: A description of the null phenotypes was present in the 2nd paragraph of the Introduction: “Null alleles of *Mlh1* or *Mlh3* cause meiotic arrest and sterility in mice because they are needed for ~90% of all crossovers (those known as interference-dependent Class I crossovers) (Eaker et al. 2002; Edelmann et al. 1996; Lipkin et al. 2002; de Boer et al. 2006; Toledo et al. 2019; Kolas et al. 2005; Anderson et al. 1999). The absence of COs leads to univalent chromosomes that fail to align properly while attached to microtubules at the metaphase plate, thus triggering the spindle assembly checkpoint (SAC) and blocking anaphase progression (Tachibana-Konwalski et al. 2013).” Nevertheless, we have added reminders of these phenotypes in relevant sections of the Results. In particular, we added that the testis histological phenotype of the severe *Mlh3* allele R1230H was “reminiscent of null animals for both *Mlh1* and *Mlh3*” and prefaced the beginning of the subsequent paragraph concerning metaphase arrest with, “Lack of COs between homologs at meiotic metaphase I in *Mlh*-null mutants causes a failure of proper alignment at the metaphase plate, leading to anaphase block 11,20–25.”

- Mutation of K618: The K618T variant gives the strongest phenotype of all MLH1 mutants and stands out in most assays (Fig. 1-3). Yet, it is noted that this variant does not exist in the population, and it is rather present as K618A. Surprisingly, my understanding is that K618A variant does not present abnormal phenotype. How can that be? Is it possible that the most serious mutations cause embryonic lethality in humans, and only "weaker" mutations then associate with infertility?

RESPONSE: Regarding embryonic lethality, knockouts in MLH1 or MLH3 are not lethal, but can predispose to cancer (Lynch syndrome) in people, as mentioned in the Discussion. Indeed our preliminary analyses of K618A indicate that it is either an entirely neutral allele, or an allele with only subtle defects. Actually, it is not at all surprising to us that this single missense allele doesn't have a drastic (or any) effect; we have made numerous alleles in various reproductive genes that have no effect (e.g. PMID:26240362, PMID:31393579, PMID:30085085), underscoring the shortcomings of in silico prediction algorithms. Nevertheless, as the reviewer suggests, severe alleles (causing highly penetrant infertility) would be subject to negative selection and likely not attain significant levels in populations.

Reviewer #2 (Remarks to the Author):

The funding function of meiosis is to keep homologs joined to allow chromosome reduction. Genetic variation at the essential mismatch repair genes MLH1 and MLH3 might potentially be responsible of human infertility and / or RPL and miscarriages. Based on this hypothesis, the authors evaluated human missense SNPs curated in gnomAD databases. By in silico modelling

and biochemical assays / yeast genetic assays the authors selected those more severe variants for modeling in mice. The results show that the mouse models exhibited crossing over defects that led to age-related decreases in female fecundity, elevated aneuploidy, and consequent increases in pregnancy loss. The work is very well designed and the analysis is well conducted. The results are of great relevance in the field of human biology and reproduction and demonstrate that human genetic variants (relatively common) in the meiotic recombination machinery can lead to human infertility and miscarriages. Interestingly, these genetic variants in females led to age-dependent subfertility and aneuploidy. This observation is novel and of great relevance and provide a novel hypothesis to explain age-dependent RPL and trisomies such as Down syndrome. The MS is thus of great interest and relevance for the scientific community.

RESPONSE: Thanks for crystallizing the key impacts of the paper!

I have some few minor comments:

- It would be of help for understanding the Y2H disruption section to incorporate in the Result section a brief explanation of how / why those genes/proteins were selected for this analysis (beyond mat and methods).

RESPONSE: Thanks for the suggestion. We added a sentence in Results explaining that the genes selected were based on the fact that they were already known to interact with MLH1, and that full length human clones were already available in validated libraries.

- From my perspective, I miss a rationale explanation for the different pipeline used for the selection of the MLH1 and the MLH3 variants. This explanation would clarify why this has been done in such a way.

RESPONSE: We now include a sentence at the end of the first paragraph of Results explaining that as we and many other have previously noted, in silico functional prediction algorithms are not sufficiently reliable for identifying phenotypic mutations, so we aimed to augment the performance of such algorithms by coupling their predictions with the results of in vitro functional assays to better prioritize candidate mutations for modeling in mice. To accomplish this, we required functional assays that were both (1) compatible with our genes of interest (MLH1 and MLH3), and (2) amenable to relatively high throughput study since so many candidate variants exist. No single in vitro assay could span the numerous proteins involved in meiotic recombination nor is there any guarantee of the translatability of any single assay's results to phenotypic mutations in mice. As such, we leveraged ongoing collaborations with the Alani and Yu Labs here at Cornell whom have expertise in the meiotic crossover and protein-protein interaction assays presented here for MLH3 and MLH1, respectively.

- The images of the chromosome analysis carried out (figure 3F) to determine the abnormal number of chromatids at MII are not of good quality in my electronic version of the MS. From this figure it is difficult to figure out what is happening at those plates. It should be needed a clearer demonstrative figure.

RESPONSE: In a new Figure 3F, we made modifications to increase the image quality. In addition, we add numbers to identify each pair of sister chromatids and insets with the corresponding pairs. This presentation allows for ease of viewing the pairs while highlighting the mutants with abnormal pair numbers, and premature sister separation (using a red outline).

-Interestingly, despite the absence of reproduction defects in the 6 male mice mutants, histological analysis revealed the presence of abnormal metaphases. In this respect, the confocal analysis carried out does not allow to show in detailed the chromosome segregation defects (S3). Immuno Fluorescence analysis of metaphase I from squashed tubules using for instance SYCP3 and DAPI would notably increase the detailed and resolution of the analysis (are XY more frequently involved?).

RESPONSE: Because most of the mutants had little or no effect on fertility or sperm numbers, and the fraction of tubule sections with at least 1 TUNEL+ cell was low (10-20%), and metaphase is very short and thus rare, quantifying the magnitude and spectrum of possible recurrent XY disjunction would be difficult and not really central to the female (oocyte) emphasis of the paper. If there were specific autosomes being affected preferentially, the "N" problem would be compounded, and FISH would have to be applied in addition. In short, though the suggested experiments could give more clarity about the general extent of disruption in abnormal metaphases, we don't believe it is crucial for the main points of the paper. We note however, that we are embarking on a project to look at embryo aneuploidy resulting from mutant oocytes (from mutants with varying CO numbers), at different ages, using genomic analysis. We hope to have data in a few years on the relationship between CO number, age, and aneuploidy rates/spectrum.

-Female analysis for fertility would be improved if it would follow the same script structure used in the male analysis such as the MLH1 staining of COSs. This analysis (CO counts in oocytes) would enable to demonstrate (similarly as it has been shown in males) that COs are reduced or absent and would allow to compare the severity of the phenotypes of each variant in males vs females, which is also of great interest given the known sexual dimorphism of meiotic recombination.

RESPONSE: We agree that this data would have been a nice addition to the paper, but we emphasized studies of chiasmata, the ultimate outcome of COs (MLH1 foci are an indirect measure) to test the idea that reduced litter sizes were a consequence of aneuploidy driven by reduced COs. Nevertheless, in future work for which we have recently obtained funding, we are planning to conduct a comprehensive study of these mutants to determine if there is a correlation between maternal age and actual CO number in oocytes. However, this is beyond the scope of the report here.

Reviewer #3 (Remarks to the Author):

In the manuscript titled “MLH1/3 variants causing aneuploidy, pregnancy loss, and premature reproductive aging” by Singh et al., the authors model and study human variants of MLH1/3 in mice. These mismatch repair genes are required for crossovers during meiosis and null alleles cause sterility in mice. The authors perform a rigorous selection procedure involving a combination of computational and molecular assays to choose human population variants of MLH1/3 and then perform a detailed analysis of their functional impact on male and female fertility in mice. They describe the molecular defects caused by the Mlh1 and Mlh3 mutations during male meiosis. In females Mlh1 mutations cause an increase in aneuploidy in oocytes resulting in non-viable embryos. Interestingly, this manifests as an age-related decline in female fertility and the authors discuss the implications of these findings. This is a heroic, unique and thorough study. The manuscript is in good shape and suitable for publication in Nature Communications as is. Few minor comments are listed below.

RESPONSE: Thanks for the kind comments. Indeed this took many years of work.

Minor comments:

1) The schematic for MLH1 variant selection in Figure 1 is hard to follow – please clarify/correct. The text describes selection of variants with $AF < 2\%$ but the figure shows different selection criteria ($< 1\%$) for MAF (which I’m assuming is mutation allele frequency). Also the text states that 13 MLH1 variants were tested by Y2H and then those that both disrupted interactions and scored poorly in functional predictions algorithms were modeled, but the pathogenicity algorithms filter is listed at an earlier step in the figure. Also the figure shows 20 SNPs being tested, whereas the text states that 13 SNPs were tested.

RESPONSE: Thanks for noting the inconsistencies. Some edits in Fig. 1 and Table S1 didn’t make it into the submitted version, and we have now corrected them. We changed MAF (Minor allele frequency) to AF (allele frequency) in the figure and defined it in the legend. The AF range we used is corrected to 0.02% - 2%, and variants outside this range were removed from Table S1. We switched the order of AF and pathogenicity prediction algorithms in Fig. 1A. The figure and text and Table S1 were harmonized to show 13 SNPs tested by Y2H.

2) The text on Page 5 states that MLH3-R1230H mutants show prophase arrest. Do the authors mean metaphase arrest?

RESPONSE: Thanks for pointing out! Corrected to metaphase.

3) I think a few sentences describing the histological phenotypes in Figure 1(D) in greater depth would be useful for the reader. At least for the two alleles with strong phenotypes that are described in more detail: MLH3-R1230H and MLH1-K618T.

RESPONSE: We added a sentence in the histological description for each of these two alleles, and also modified Fig. 1D and its legend to distinguish between pyknotic spermatocytes and spermatocytes with abnormal metaphase plates or lagging chromosomes.

4) Figure 2(D) legend: state what the arrows annotate.

RESPONSE: Addressed in previous question.

5) Figure 2(E) is not referenced in the text. And text states that there is a 2-fold reduction in chiasmata in K618T but the figure shows a small median reduction per cell - please clarify.

RESPONSE: Thanks for noticing the errors. We added the Fig. 2E reference, and also deleted "fold" from the text. The corrected version reads "*Mlh1*^{K618T/K618T} spermatocytes exhibited an average of ~2 fewer metaphase I chiasmata than controls."

6) The authors state that the *Mlh1* mutant females cease to have litters at a younger age compared to wild type despite continued ovulation. A sentence describing the actual data in the text would be nice. How much earlier did the females cease to have litters? Was this phenotype approximately the same for all mutants?

RESPONSE: To enhance the clarity of the results, we have now added a few sentences describing the actual data in the text : "The overall shortfall became dramatic when mutant dams were 5-6 months old and was largely attributable to the following: **i**) decrease in fecundity after the first ~six litters (Fig. 3A); **ii**) longer intervals between litters; and **iii**) earlier cessation of litters, despite evidence for continued ovulation (Fig. S5). Collectively, these resulted into 1.6-6.5 fold fewer pups from all 5 mutant dams compared to WT. The most severe phenotype were displayed by *Mlh1*^{K618T} and *Mlh1*^{V716M} dams. After their first five litters, these dams produced 2.2-fold and 6.5-fold fewer pups than WT, respectively, (22±6.1; 7.7±2.6; 49.8±8 SEM)."

7) Figure 3(E) incorrectly references (A), I think the authors meant (D).

RESPONSE: Thanks for noting the error; it has been corrected.

8) For Figure 3(F, G), it would be useful to have a more detailed description of precisely what staining was scored as aneuploid and a mention of the patterns seen (description of the insets) in the text.

RESPONSE: Thanks for the helpful advice. As mentioned in the response to Reviewer 2, Figure 3F was revised to add numbers to identify each pair of sister chromatids and insets with the corresponding pairs. This presentation allows for ease of viewing the pairs while highlighting the mutants with abnormal pair numbers, and premature sister separation (using a red outline). The revised version of the text reads: "Next, we evaluated the ability of these females to produce euploid eggs (20 pairs of sister chromatids), and then we quantified the number of sister chromatid pairs in eggs from each mutant. The *Mlh1*^{E268G}, *Mlh1*^{K618T} and *Mlh1*^{V326A} alleles presented marked increases in aneuploid eggs, detected as abnormal number of pairs of sister chromatids (Fig. 3F and insets, G; 20.9±7.1%, 25.8±12.9% and 12.7±2.3, respectively, compared to 4.5±2.7% in WT). Moreover, eggs from *Mlh1*^{E268G} and *Mlh1*^{V326A} alleles presented an abnormal phenotype of prematurely separated sister chromatids (Fig. 3F insets, outlined red)."

9) In Figure 3(H) the whiskers do not seem to represent the maximum and minimum values as stated in the legend.

RESPONSE: Thank you for pointing out; the legend was edited to describe them.

10) I suggest coloring the THR1 locus in Figure S1 the same blue as the blue spores to avoid confusion with purple spores.

RESPONSE: Thanks; done.

REVIEWERS' COMMENTS

Reviewer #1 (Remarks to the Author):

The authors answered all my queries. The study is of a very high quality, and I will look forward to seeing it published.

Reviewer #2 (Remarks to the Author):

The authors have addressed satisfactorily all the minor points previously mentioned in my report. Consequently, the revised MS should be considered for its direct acceptance in Nature Communications.

Alberto M Pendas

Reviewer #3 (Remarks to the Author):

The authors have addressed all my comments and I have no additional comments.

Response to Reviewers

The last review had no requests for edits, and the manuscript has been provisionally accepted. This submission contains technical edits as requested by the editor and to meet formatting and reporting requirements. Also to submit high def image files.